# The Relationship between Electron Transport and Microstructure in Ge_2_Sb_2_Te_5_ Alloy

**DOI:** 10.3390/nano13030582

**Published:** 2023-01-31

**Authors:** Cheng Liu, Yonghui Zheng, Tianjiao Xin, Yunzhe Zheng, Rui Wang, Yan Cheng

**Affiliations:** 1Key Laboratory of Polar Materials and Devices (MOE), Department of Electronics, East China Normal University, Shanghai 200241, China; 2Chongqing Key Laboratory of Precision Optics, Chongqing Institute of East China Normal University, Chongqing 401120, China

**Keywords:** phase change memory, Ge_2_Sb_2_Te_5_, phase transition, electron transport

## Abstract

Phase-change random-access memory (PCRAM) holds great promise for next-generation information storage applications. As a mature phase change material, Ge_2_Sb_2_Te_5_ alloy (GST) relies on the distinct electrical properties of different states to achieve information storage, but there are relatively few studies on the relationship between electron transport and microstructure. In this work, we found that the first resistance dropping in GST film is related to the increase of carrier concentration, in which the atomic bonding environment changes substantially during the crystallization process. The second resistance dropping is related to the increase of carrier mobility. Besides, during the cubic to the hexagonal phase transition, the nanograins grow significantly from ~50 nm to ~300 nm, which reduces the carrier scattering effect. Our study lays the foundation for precisely controlling the storage states of GST-based PCRAM devices.

## 1. Introduction

Phase-change random-access memory (PCRAM) is one of the most promising and mature new memory technologies due to its high operating speed, cycle life comparable to that of dynamic random-access memory, and large operation window [1,2,3]. As the core of PCRAM, phase-change materials (PCMs) store information by the huge difference in resistivity between amorphous (high resistance) and crystalline state (low resistance) [4,5]. Currently, the Ge_2_Sb_2_Te_5_ (GST) alloy is one of the most mature PCMs due to its excellent electrical properties, and it is also the parent phase for constructing low-power superlattice or the doped material [6,7]. There are two crystalline structures in GST materials: one is the metastable face-centered cubic (f-), and the other is the stable hexagonal (h-) structure [8,9]. In the f-phase, Te atoms occupy the anion lattice, and the Ge/Sb/20% vacancies occupy the cation lattice. With vacancy aggregation, the f-phase will transform into the h-phase at high temperatures [10,11].

Although literature investigates the microscopic storage mechanism of GST from experiments and theoretical methods, the relationship between the electronic transport property and the microstructure is not fully elucidated currently [12,13,14,15]. In GeSb_2_Te_4_, a similar alloy to GST, electronic measurement showed that it undergoes a metal–insulator transition mechanism, accompanied by the disorder-to-order process [16,17], while the corresponding structure variation is not well explained.

In this work, we investigated the relationship between the electric property and microstructure through resistance–temperature (R-T) curves, in-situ transmission electron microscope (TEM) heating, Hall test, and the electron backscattering diffraction (EBSD) technique. With the increasing temperature, the amorphous GST film first transformed into the f-phase, and then the h-phase, accompanied by the two sharp resistance drops. At the first drop, the carrier concentration increased substantially due to the variation of the atomic bonding environment. At the second drop, the nanograins grew significantly with an obvious [0001] texture, increasing the carrier mobility.

## 2. Materials and Methods

### 2.1. Sample Preparation

GST films (~200 nm thick) were deposited on a Si (100)/SiO_2_ substrate by a magnetron sputtering stoichiometric Ge_2_Sb_2_Te_5_ target in magnetron sputtering equipment (ACS-4000-C4, ULVAC, Kanagawa, Japan), and the films for TEM observation were deposited on TEM grids with a thickness of 30 nm. The base pressure of the vacuum system is about 1.5 × 10^−4^ Pa, the Ar flow rate is 50 sccm, the sputtering DC power is 50 W and the corresponding sputtering pressure is about 0.2 Pa. With the aid of energy dispersive spectroscopy (EDS) accessory (EDAX, Oxford, UK), the average concentrations of Ge, Sb, and Te atoms were estimated to be 20.7%, 29.4%, and 49.9%, respectively. Then an amorphous SiO_2_ film (~5 nm in thickness) was deposited on top of GST films as the anti-oxidation layer.

### 2.2. The Hall Test

The electrical property of the film was obtained by the Hall measurement system (H-50, MMR, Baton Rouge, LA, USA) at room temperature. First of all, the sample film was cut into small squares (1 × 1 cm^2^ to 1.3 × 1.3 cm^2^). Each sample was annealed in Ar gas at different temperatures. The annealing temperature range was from 100 °C to 300 °C, and the interval step was 25 °C. Each sample was tested 20 times for accuracy.

### 2.3. Microscopic Characterization

The microstructure of the film was studied using X-ray diffraction (XRD) in an X-ray diffractometer (PANalytical X’Pert PRO, PANalytical B.V., Almelo, The Netherlands) with Cu Kα radiation source. The in-situ crystallization process of the GST film was carried out in a Gatan 628 heating holder (Gatan 628, Gatan, Pleasanton, CA, USA) with a heating rate of 20 °C/min. The bright field (BF) images and selected area electron diffraction (SAED) patterns were obtained in JEOL 2100F TEM (JEM-2100F, JEOL, Tokyo, Japan) under 200 kV. The phase structure at 180 °C and 270 °C was obtained in the focused ion beam system (Helios G4 UX, FEI, Hillsboro, OR, USA) using EBSD technology. The voltage was 30 kV and the step size was 10 nm and 50 nm, respectively.

## 3. Results

Figure 1a shows the resistance–temperature (R-T) curves of GST films annealed for different temperatures (100–300 °C). These samples went through a constant heating process with a rate of 5 °C/min from room temperature by the two-probe method. Before 150 °C, the resistance gradually decreased from 10^9^ Ω to 10^6^ Ω. Around 150 °C, a sharp resistance decrease could be observed (from 10^6^ Ω to 10^4^ Ω), corresponding to the amorphous-to-f phase transition. Further increasing the temperature to 250 °C, the resistance continued to decrease to 10^3^ Ω. Around 250 °C, the second resistance decrease could be observed (from 10^3^ Ω to 10^4^ Ω), corresponding to the f-to-h phase transition. The structure of GST films at different temperatures was also investigated through XRD experiments. The results shown in Figure 1b confirmed an amorphous state at room temperature, the f-phase at 200 °C, and the h-phase at 300 °C.

The R-T curves during the cooling process are also investigated to explore the metal–insulator transition in GST films. When the annealing temperature was lower than 150 °C, the resistance of GST films backtracked to 10^9^ Ω at room temperature. When the films crystallized into the f-phase, the cooling R-T curves no longer coincided with the heating R-T curve, and the temperature coefficient of resistance (TCR, yellow lines) was less than zero, demonstrating a semiconductor behavior. The higher the annealing temperature, the larger the TCR is. At 250 °C, the TCR is ~0.64, indicating that metal–insulator transition (MIT) occurs.

To explore the electronic transport property in GST films, Hall experiments were conducted. Figure 2a shows the schematic of the Hall test [18], when the current, magnetic induction, and sample thickness are known, it is only necessary to measure the Hall voltage to obtain the carrier concentration and mobility. Here, we did not use the in-situ heating Hall test method because of a large experiment error. Instead, we carried out the Hall test with squared samples directly. After annealing, the structure of the sample was stable, and the obtained Hall effect parameters were relatively accurate, which reflects the relationship between the electronic transport properties and the structure. Here, the films annealed from 25 °C to 350 °C with an interval step of 25 °C were investigated. Figure 2b shows that the resistivity was ~5 × 10^3^ Ω·cm at low temperatures. When the temperature rose to 150 °C, the resistivity sharply decreased to 10^0^ Ω·cm, corresponding to the amorphous-f phase transition. As the annealing temperature increased to 225 °C, the resistivity dropped to 10^−3^ Ω·cm, corresponding to the f-to-h phase transition. The mismatch of the f-to-h phase transition temperature in Figure 1a and Figure 2b should be ascribed to the two different thermal treatments, while both of them maintained a low-level resistivity at higher temperatures. Figure 2c,d shows the carrier concentration and carrier mobility of the film at different temperatures. Before 150 °C, the carrier concentration was ~5 × 10^15^ Ω·cm^−3^. At 150 °C, the carrier concentration sharply increased to ~1 × 10^19^ Ω·cm^−3^, which shows the same variation orders of the magnitude observed in Figure 1a. Therefore, the decrease in resistance during crystallization is mainly attributed to the rapid increase in carrier concentration. When the temperature continued to increase, the carrier concentration was maintained at a high level. Interestingly, the carrier mobility suddenly increased from 1 cm^2^/(V·s) to ~50 cm^2^/(V·s) at 225 °C; thus, the decrease in resistance during the f-to-h phase transition is related to the increase of carrier mobility. At high temperatures, the value maintained around ~10 cm^2^/(V·s) to ~100 cm^2^/(V·s).

To understand the electronic transport property from a microscopic point of view, an in-situ heating experiment of GST film inside TEM was performed as shown in Figure 3. At room temperature, the morphology of the film was uniform, and the corresponding SAED pattern showed a diffused ring (Figure 3a), demonstrating a typical amorphous state. At 140 °C, some nanograins appeared and (220) a lattice plane (a continued and sharp ring) of the f-phase was detected, demonstrating the start of crystallization. As the temperature rose to 240 °C, the nanograins and other f-phase lattice planes were more and more obvious, but note that the sizes of the nanograins were still small. Therefore, the increase of carrier concentration is mainly related to amorphous to f-phase transition. During the crystallization process, the tetrahedral Ge atoms jumped to the octahedral coordination, and the change in the constituent atom environment increased the carrier concentration. With increasing temperature, the vacancies gradually aggregated, while the variation of carrier concentration or carrier mobility was small. At 278 °C, a large grain suddenly occupied the entire top left area (bright contrast, Figure 3e). As the temperature increased to 300 °C, the grain quickly merged the nanograins throughout the whole area as shown in Figure 3f,g [19]. The corresponding SAED patterns (Figure 3) showed a typical single crystal structure, which belongs to the [0001] orientation of the h-phase.

Through an in-situ heating experiment inside TEM, we observed the phase transition process in a small area. To study the crystal structure in a larger area, we used the EBSD technique to analyze the GST film that was annealed at 180 °C and 270 °C, respectively. Figure 4a,c show the inverse polar figure (IPF) of the film at 180 °C and 270 °C, respectively. At 180 °C, the orientation difference of the IPF was very inconsistent, while the orientation of the grains was very similar at 270 °C, demonstrating that the film had a preferred orientation, which corresponds to the [0001] orientation in the h-phase as observed in Figure 3h. Further, the statistical grain size distribution [20,21,22] at two temperatures are also shown in Figure 4b,d. The calculated average grain size was 48.5 nm for 180 °C, and 292.3 nm for 270 °C, indicating that the nanograins grew significantly during the f-to-h phase transition.

To explore the preferred orientation relationship between the f-phase and the h-phase, we analyzed the EBSD pole figures (PF) for {100}, {110}, and {111} crystallographic planes for the f-phase (180 °C). As can be seen from Figure 5a, the maximum density of orientation difference was 2.97; thus, there was no consistent preferred orientation in the f-phase films at 180 °C. As for the PF in the h-phase films (270 °C), there was no preferred orientation in {11$\bar{2}$0}, and {10$\bar{1}$0} planes, while in the {0001} plane, the maximum orientation difference density was 65.15 at the central position, which is along the [0001] orientation. Therefore, the large h-phase grains in GST film indeed had a preferred orientation as observed in Figure 3h. During the f-to-h phase transition, the carrier concentration did not increase significantly. However, due to the growth of the nanograins, the scattering affected by the crystal lattice was weakened, hence increasing carrier mobility. When the structural transformation was completed, the grain size did not change, and thus the mobility reached a stable value at this time.

## 4. Conclusions

In summary, the relationship between electronic transport property and microstructure in GST alloy was studied. Induced by thermal treatment, GST alloy first crystallized into the f-phase, with a sharp resistance drop. At the same time, the carrier concentration increased substantially due to the variation of the atomic bonding environment during the crystallization process. At higher temperatures, the carrier concentration maintained a high level. When the temperature was high enough, a second resistance drop was observed, which is related to the increase in carrier mobility. This is because the f-phase grains transformed into the h-phase, and significantly grew up from ~50 nm to ~300 nm, which could reduce the carrier scattering effect. Our study lays the foundation for establishing the relationship between electron transport and microstructure in GST materials, and also provides an optimization direction for precisely controlling the storage states of GST-based PCRAM devices.

## Figures and Tables

**Figure 1 nanomaterials-13-00582-f001:**
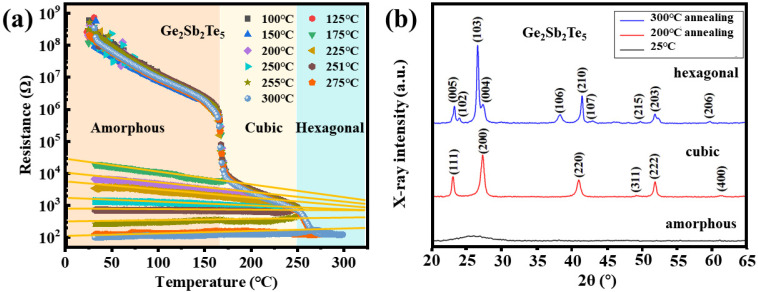
(**a**) The resistance–temperature curves of GST films annealed from 100 °C to 300 °C. (**b**) XRD patterns at 25 °C, 200 °C, and 300 °C, respectively.

**Figure 2 nanomaterials-13-00582-f002:**
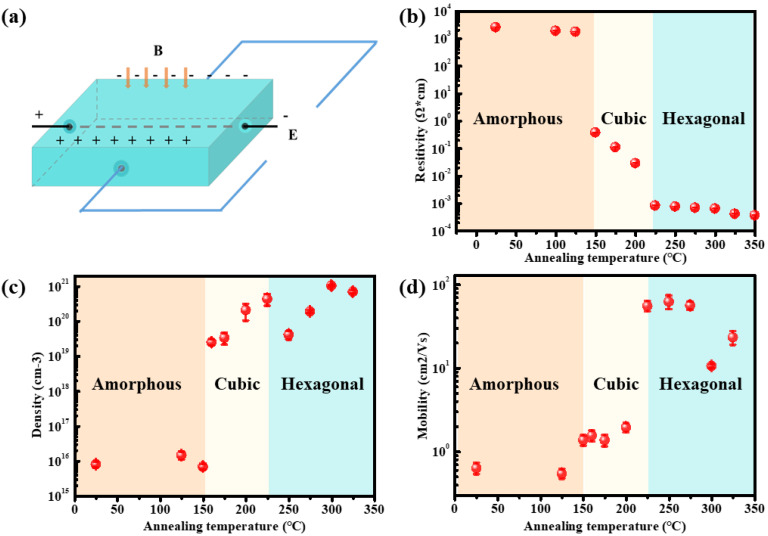
(**a**) The Hall test schematic. (**b**) The resistivity, (**c**) carrier density, and (**d**) carrier mobility at different annealing temperatures.

**Figure 3 nanomaterials-13-00582-f003:**
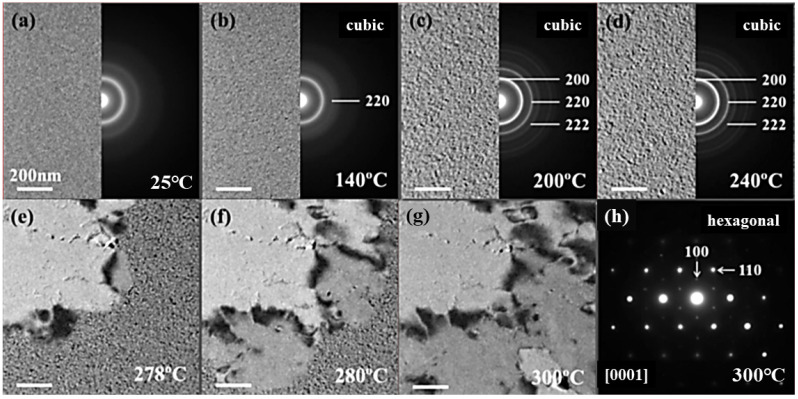
(**a**–**d**) The morphology and SAED patterns of GST films at 25 °C, 140 °C, 160 °C and 200 °C, respectively. (**e**–**g**) The morphology of GST films at 278 °C, 280 °C, and 300 °C, respectively. (**h**) The SAED patterns at 300 °C, which is along the [0001] axis.

**Figure 4 nanomaterials-13-00582-f004:**
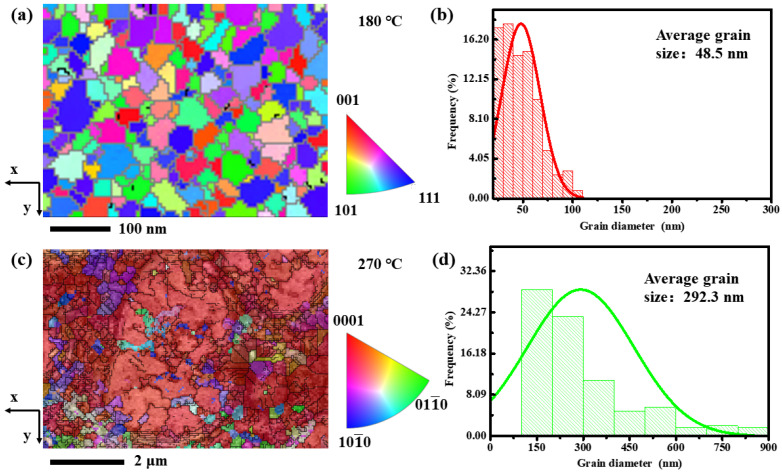
(**a**,**b**) are the inverse polar figure (IPF) of the film and the statistical grain size distribution at 180 °C. (**c**,**d**) are the IPF of the film and the statistical grain size distribution at 270 °C.

**Figure 5 nanomaterials-13-00582-f005:**
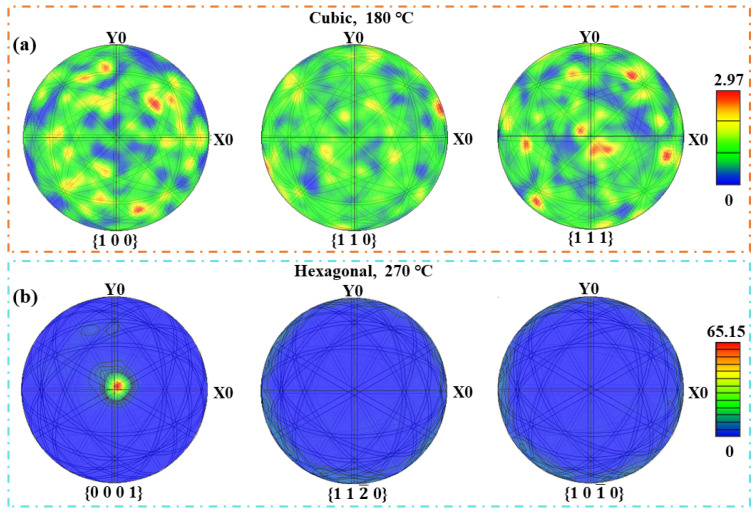
(**a**) EBSD pole figures for {100}, {110}, and {111} crystallographic planes for the f-phases film (180 °C). (**b**) EBSD pole figures for {0001}, {11$\bar{2}$0}, and {10$\bar{1}$0} crystallographic planes for the h-phase films (270 °C).

## Data Availability

All data that support the findings of this study are available from the corresponding authors upon reasonable request.

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
