# Peer review of "The Relationship between Electron Transport and Microstructure in Ge_2_Sb_2_Te_5_ Alloy"

_nanomaterials, 2023, doi:10.3390/nano13030582_

Round 1
Reviewer 1 Report
This manuscript reports a comprehensive study of the electrical properties and microstructures of GST across the phase transitions. It is of some interest to the field and should be published with revision. I have a few questions and comments below.
1. This is not the first research on the Hall effect measurement of GST, the authors need to compare their results with prior work like Journal of Applied Physics 110, 013703 (2011) and many other experimental and theoretical work on this topic. It looks like in this early JAP paper, the carrier concentration change is not observed in GST phase transition.
2. The information about the film growth is not adequately described. What is the target material? What is the sputtering power? Is this magnetron sputtering?
3. There is a lack of the confirmation of the film composition. Additional characterization using methods like XRF, Ruthford back scattering, or EDX are needed to demonstrate if the samples stoichiometry to complement the XRD measurements.
4. Please clarify that the Hall effect measurement is carried out at room temperature. What is the type of carrier? It looks like electron is the carrier but it is not mentioned in the text. Did the author exchange the direction of the magnetic field from top down to bottom up to exclude the magnetoresistance and other factors in addition to Hall effect that can contribute to the transverse valtage?
5. What is the method used in the longitudinal resistance measuremnet? Did they use colinear 4-probe or van der Pauw method?
6. The temperature range of the Hall measurement mentioned in the text (100-300 C) is not consistent with the figure (125-325C).
7. The resolution of Fig.4c is too low to show the grain size. Since the grain size of the sample increase from ~50 to ~300 nm, it is not convincing to state that the grain orientation change is the dominant reason for the resistance change. Please rephrase the corresponding sentence in the abstract and the occlusion to avoid misleading information.
8. The English of the manuscript must be improved with numerous grammar issues. For example in the abstract the sentence "As the matures phase change materials, Ge2Sb2Te5 alloy (GST), it mainly relies on the electrical differences between different states to realize information storage,..." Should be "As a mature phase change material, Ge2Sb2Te5 alloy (GST) mainly relies on the distinct electrical properties between different states to realize information storage.,...". The article "the" is missing in front of almost all cases when they mention "f-phase" or "h-phase". In line 40, the word "material" is redundant. The tense of the sentence in line 47 is wrong or nor consistent.
Reviewer 2 Report
The manuscript "The relationship between electron transport and microstructure 2 in Ge2Sb2Te5 alloy" by Zheng et al. summarizes experimental studies of the property changes in the sputter deposited phase-change material Ge2Sb2Te5 (GST) upon annealing and structural relaxation. The electrical resistance is most relevant for application. The authors attribute its increase upon crystallization and transition to the stable hexagonal to an increase in carrier concentration and an increase in carrier mobility, respectively. While the experiments are very similar to those reported in ref. 15 for the PCM GeSb2Te4, there is a key difference in the interpretation: Siegrist et al. concluded that a metal-insulator transition (MIT) occurs within the cubic phase (called f-phase here), i.e., independent from the f-to-h transition. Here, the MIT is found to coincide with the f-to-h transition. Unfortunately, the experimental evidence in this manuscript is insufficient to support this claim, because the structural information (via X-ray diffraction, XRD) was obtained using a different heating and annealing protocol than the electric measurements determining the MIT. Why did the authors not perform XRD measurements on the annealed samples used for the resistance and Hall-measurements at ambient conditions? This drawback of the present manuscript renders it insufficient to support publication in nanomaterials. There is, however, the chance that the authors can compensate this drawback in a revised version. There are a few more concerns, which should be addressed in a revised version. This mostly concerns the description of the experiments, where important details are missing, see below. In a revised version, also the use of English language should be optimized in sentences like "while corresponding structure variation is not very understood".
Minor concerns:
1) p. 2, l. 56: A thickness of 200 nm is given, but does this refer to the PCM or to the TEM-grid? If it refers to the TEM grid, were all studies performed on 200 nm thick films grown under identical conditions?
2) Were the samples not capped with a passivation layer to prevent oxidation of the PCM? This needs to be explicitly clarified as different protocols are in common use and the passivation layers affect the phase transition temperatures.
3) p. 2, l. 64: "Each sample was tested 20 times.": What does this mean? Was the sample installed 20 times in the test setup to perform independent measurements? If so, why are not error bars included in the data in Fig. 2 that could clarify the need for doing this measurement 20 times.
4) In all figures where the annealing temperature is shown rather than the actual temperature (e.g. Fig. 2, panels b-d), the label "temperature" is misleading and should be replaced by "annealing temperature".
Round 2
Reviewer 2 Report
All concerns from my previous report were resolved by the authors' response and the changes made in the manuscript, which is recommended for publication in nanomaterials in its present form.
